# Discovery of 2′,6-Bis(4-hydroxybenzyl)-2-acetylcyclohexanone, a Novel FtsZ Inhibitor

**DOI:** 10.3390/molecules27206993

**Published:** 2022-10-18

**Authors:** Hsuan-Yu J. Lin, Rachana Rao Battaje, Jinlong Tan, Munikumar Doddareddy, Hemendra Pal Singh Dhaked, Shalini Srivastava, Bryson A. Hawkins, Laith Mohammad Hilal Al-Shdifat, David E. Hibbs, Dulal Panda, Paul W. Groundwater

**Affiliations:** 1Sydney Pharmacy School, Faculty of Medicine and Health, The University of Sydney, Sydney, NSW 2006, Australia; 2Department of Biosciences and Bioengineering, Indian Institute of Technology Bombay, Mumbai 400076, India; 3National Institute of Pharmaceutical Education and Research, Nagar 160062, India

**Keywords:** cell division, Z-ring, FtsZ inhibitor, chemical synthesis, molecular docking

## Abstract

Multi-drug resistance is increasing in the pathogenic bacterium *S. pneumoniae*, which is mainly responsible for meningitis and community-acquired pneumonia (CAP), highlighting the need for new anti-pneumococcal agents. We have identified a potential anti-pneumococcal agent, enol **3**, which acts by hindering the cell division process by perturbing Z-ring dynamics inside the cell. Enol **3** was also shown to inhibit FtsZ polymerization and induce its aggregation in vitro but does not affect the activity of tubulin and alkaline phosphatase. Docking studies show that **3** binds near the T7 loop, which is the catalytic site of FtsZ. Similar effects on Z-ring and FtsZ assembly were observed in *B. subtilis*, indicating that **3** could be a broad-spectrum anti-bacterial agent useful in targeting Gram-positive bacteria. In conclusion, compound **3** shows strong anti-pneumococcal activity, prompting further pre-clinical studies to explore its potential.

## 1. Introduction

The bacterial cell-division pathway has been closely studied as a target for new antibacterial agents [1], as numerous studies carried out in *E. coli* and *B. subtilis* have resulted in a greater understanding of bacterial cell division [2,3,4]. These studies have characterised a group of conserved proteins which are systematically assembled in order for the cell to undergo cytokinesis. A key protein, and a potential drug target, in bacterial cytokinesis is filamenting temperature-sensitive mutant Z (FtsZ); the existence of FtsZ was discovered through mutational studies in temperature-sensitive *E. coli* in the 1970s, in which *E. coli* strains were found to have septum formation defects at 42 °C that led to elongated filaments rather than normal viable cells [5,6]. Despite it having been validated as a target for antibacterial agents [7,8,9], no FtsZ inhibitor has yet successfully negotiated phase I clinical trials.

We have recently reported that a non-natural curcumin analogue **1** possesses powerful antibacterial activity against important pathogenic bacteria, as a result of its targeting of FtsZ [10]. Building upon this discovery, we now wish to report on the inhibitory activity of dihydrochalcone **2** and reduced enol **3** analogues. The rationale behind the development of these compounds was to remove the functional groups in bis(enone) **1** deemed to be potentially problematic in terms of possible sites of degradation/metabolism, while hopefully maintaining the desired antibacterial activity.

There has been significant debate regarding the value of curcuminoids as lead compounds in drug discovery as, in addition to its poor pharmacokinetic profile, curcumin has contentiously been categorised among the PAINs (pan-assay interference compounds); PAINs are compounds which show deceptively attractive in vitro results in a variety of biological assays (typically in high throughput screens) due to mechanisms unrelated to the target of the assay [11,12,13]. Knowledge of the structural contributors to the curcumin’s apparent PAINs behaviour of curcuminoids allows for the rational design of structural analogues aimed at removing or modifying such moieties, whilst hopefully retaining the desired biological activity. 

One of the aims of this work was to address possible PAINs-related issues in bis(enone) **1**, through the reduction of the enone groups to give the 1,3-diketone **3**, thereby removing the possibility of covalent adduct formation involving Michael attack on the β-carbon atom. A transfer hydrogenation method which had previously been used to reduce curcumin to a tetrahydrocurcumin, using palladium as the catalyst [14], was initially trialled to reduce both enones. Another target was the phenol product **2** of the disproportionation of diketone **1** (Figure 1).

## 2. Results and Discussion

### 2.1. Chemistry

Bis(enone) **1** was prepared as described previously, and the X-ray diffraction of a single crystal allowed the identification of the enol form **1′** with the hydroxyl group on the cyclohexyl ring (Figure 2); the O1-C1 bond length was determined to be 133.4 pm, while O2-C7 was 127.8 pm, both of which correlate well with established bond lengths, specifically 133.3 pm for enol C=**C-O**H bonds and 122.2 pm for C=C-**C=O** ketone bonds [15].

In a process described by Eschinazi and Bergmann, *d*-limonene undergoes a disproportionation in the presence of a palladium catalyst to give *p*-cymene, an aromatic compound, and *p*-menthane, a saturated cycloalkane [17]. We postulated that the transfer of hydrogens, which occurs during the disproportionation, could act as reducing equivalent for the enones and that limonene could also oxidise the cyclohexyl ring of the bis(enone) **1** by accepting the hydrogens transferred. When bis(enone) **1** was reacted with excess *d*-limonene in excess acting as a co-reactant and the carrier solvent, the phenolic derivative **2** was produced in good yield (64%). The use of cyclohexene as a hydrogen donor in catalytic transfer hydrogenation reactions is well-established, whereby cyclohexene is oxidised to benzene [18,19] and the reduction of the cyclohexyl-containing bis(enone) **1** gave, after chromatographic purification, the reduced form **3**.

### 2.2. Biological Evaluation

#### 2.2.1. Minimum Inhibitory Concentration (MIC) Testing of Phenol **2** and Cyclohexyl Enol **3** in *Bacillus Subtilis* and *Streptococcus Pneumoniae*

On initial screening, phenol **2** was shown to be a weak inhibitor of the growth of *B. subtilis* (Bs168); treatment with 15 µM **2** resulted in 21.5 ± 0.5% inhibition. The MIC for enol **3** was found to be 11 µM with an IC_50_ of 4.97 ± 0.01 µM (Figure 3A), which is higher than that of the bis(enone) **1** (IC_50_ of 1.2 ± 0.5 µM) [10].

The growth inhibitory effect of **3** was next tested in *Streptococcus pneumoniae* TIGR4 (Figure 3B), and it was found to have an MIC of 2 µM and an IC_50_ of 1 ± 0.01 µM. These values are comparable to those of antibiotics used against this pathogen, which indicates that enol **3** is a highly potent anti-pneumococcal agent. Cytotoxicity testing was also investigated, by testing in mouse skin L929 cells to obtain the concentration for 50% cell growth inhibition (GI_50_), which indicated that the inhibitory effect of enol **3** in *Streptococcus pneumoniae* TIGR4 is approximately seven times greater than its effect in the skin cells (GI_50_ = 7 ± 2.35 µM).

Although bis(enone) **1** showed greater potency than compound **3** in terms of their IC_50_ in Bs168 cells, one major drawback for **1** is its similarity to curcumin, meaning that both compounds may be classified as PAINs, which is not a consideration for enol **3**. Overall, these results indicate that enol **3** has a low MIC for clinically relevant organisms such as *Streptococcus pneumoniae*, and as it has no PAINs attributes, it has the potential to become an antibacterial hit candidate. The biological activity of this compound thus warranted further investigation, beginning with an investigation of the possible mechanisms by which it exerts its antibacterial effect.

#### 2.2.2. Compound **3** Affects the Bacterial Cell Division by Perturbing the Z-Ring, without Causing DNA or Membrane Damage

As enol **3** is an analogue of bis(enone) **1**, which affects cell division, specifically the Z-ring dynamics in *B. subtilis*, we hypothesised that it acts via a similar mechanism of action. We thus determined the effect of enol **3** on the Z-ring in *Bacillus subtilis* and *Streptococcus pneumoniae* by immunostaining against FtsZ and DNA (Figure 4). In *Bs* 168 cells, 35 ± 17% of the dividing cells had a prominent Z-ring, which reduced to 3 ± 3 % (*p* < 0.001) within 10 min of treatment with 8 μM **3** (Figure 4A). The Z-ring in *Sp* TIGR4 was also greatly affected after treatment with 2 μM **3** for 1 h (Figure 4B). The percentage of cells without a Z-ring increased to 73% in treated cells, while in the control, the percentage was 50%. In both bacteria, there was no effect on the nucleoid separation upon treatment with **3**. We further determined the effect of **3** on the membrane of the cells using the same conditions used for immunostaining of the Z-ring and nucleoids (Figure 5). By live-cell staining specific to the cell membrane with FM-4-64, we determined that the membrane of *Bs* 168 cells remains intact after 10 min of treatment with 8 μM **3** (Figure 5A). As *Sp* TIGR4 is pathogenic, we fixed the bacteria after treatment with 2 μM **3** for 1 h and imaged the morphology using Scanning Electron Microscopy (Figure 5B), observing that there were no changes in the morphology of the cells and the intact membranes in the treated cells were comparable to those in the control cells. By showing that **3** affects the Z-ring in *Sp* TIGR4 and *Bs* 168 without perturbing the nucleoid separation or membrane, we demonstrated that **3** affects the cell division process in bacteria.

#### 2.2.3. Compound **3** Binds to FtsZ with Strong Affinity

Following our study of the effect of **3** on the cells, we aimed to elucidate its effect on purified FtsZ. We started with the assessment of the binding affinity of **3** to *S. pneumoniae* FtsZ (*Spn*FtsZ). The intrinsic fluorescence of *Spn*FtsZ reduced in the presence of increasing concentrations of **3** (Figure 6A) and the binding constant (K_d_) of enol **3** to FtsZ was determined to be 2.2 ± 0.3 μM, which surpasses that of compound **1** (K_d_ = 4.0 ± 1.1 µM) [20]. K_d_ was determined by plotting the change in tryptophan fluorescence against the concentration of **3** (Figure 6B).

#### 2.2.4. Compound **3** Inhibits FtsZ Polymerization and Induces FtsZ Aggregation

We monitored the polymerization of FtsZ in the presence and absence of the compound using light scattering at 400 nm (Figure 7A). The extent of the change in the light scattering of FtsZ reduced significantly with increasing concentration of **3**, indicating that **3** inhibits FtsZ polymerization (Figure 7B). In contrast, sedimentation of FtsZ polymers formed after incubation with and without **3**, revealed that the quantity of sedimented FtsZ was only slightly reduced in the presence of **3**. The amount of FtsZ pellet formed in the presence of 10 μM **3** showed a 17% (*p* = 0.02) reduction compared to control (Figure 7C). The apparent difference between light scattering and sedimentation experiments could be due to aggregate formation. To determine whether **3** induced the aggregation of FtsZ, we visualized the effect of **3** on FtsZ assembly using transmission electron microscopy (TEM) and atomic force microscopy (AFM). FtsZ was polymerized after incubation with or without **3** and coated on the EM grids for imaging. TEM images showed the presence of evenly distributed FtsZ filaments of length 2.1 ± 1.2 μm in the case of control (Figure 7D). However, in the presence of 10 μM **3**, large aggregates were seen throughout the grid. A small number of filaments were found scattered around the grid which were 0.7 ± 0.6 μm (*p* < 0.0001) in length. Additionally, AFM images of FtsZ filaments formed in the presence of **3** also showed the presence of large aggregates which had an average height of 233.2 ± 151.2 nm (Figure 7E). The average height of FtsZ filaments in the control was 9.02 ± 5.2 nm. Apart from a few bundles with an average height of 36.9 ± 16.8 nm, no large aggregates were to be seen in the control sample. Further, we also imaged filaments of FtsZ from *Bacillus subtilis* (*Bs*FtsZ) using TEM to check whether a similar effect of **3** was seen (Appendix A). As expected, we observed large aggregates of *Bs*FtsZ in the presence of 20 μM **3**, confirming that **3** exhibits an aggregation effect on FtsZ as seen in both bacteria. Together, our results indicate that compound **3** induces aggregation of FtsZ resulting in its inhibitory action.

#### 2.2.5. Compound **3** Does Not Affect Tubulin or Alkaline Phosphatase

After determining the inhibition of FtsZ by enol **3**, we next checked the effect of **3** on the eukaryotic homolog of FtsZ, tubulin (Figure 8). We polymerized tubulin after incubation with or without **3** and observed that no significant inhibition of tubulin polymerization was seen. Similarly, we checked the effect of **3** on alkaline phosphatase, a common enzyme found in many organisms and, again, saw no detectable inhibitory activity of **3** on alkaline phosphatase. Compound **3** does not seem to exhibit non-specific interactions.

The reduction of the enone groups to give **3** was designed to remove the possibility of these groups undergoing binding to non-specific protein targets, and this is confirmed by the absence of any effect of treatment of alkaline phosphatase and the mammalian analogue of FtsZ, tubulin, with this compound.

Overall, the results from the biological testing of enol **3** have demonstrated that maintaining the carbon linker and removing the enone functionalities results in the retention of the antimicrobial effect seen for compound **1** from our previous study [10]. Against Bs168 cells, the MIC found for compound **1** was 2.0–4.0 mg/L [10], and for compound **3**, it was ≤10 µM (equivalent to 3.5 mg/L). In terms of the MIC for Bs168 cells, both compounds performed very similarly but an interesting difference can be seen in the MICs for *S. pneumoniae* between compound **3** and PC190723 **4**, the FtsZ targeting agent which has been most thoroughly investigated [20]. An MIC of ≤2 µM (equivalent to 0.7 mg/L) was determined for compound **3** and the reported MIC found for PC190723 was more than 64 µg/mL (0.18 mM), meaning that compound **3** may be superior to PC190723 against *S. pneumoniae*.



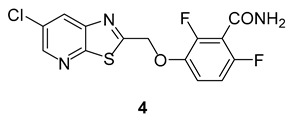



The removal of the β-diketone functional group by replacing the central cyclohexanone ring with a phenol to produce compound **2** also resulted in the retention of some antibacterial effect against Bs168.

### 2.3. Molecular Docking of Compound ***3*** in FtsZ

Since compound **3** demonstrated desirable growth inhibitory effect in Bs168 and SpnR6 cells, in silico molecular docking analysis was performed in order to demonstrate its possible binding conformation in FtsZ protein (PDB ID: 4DXD) at the co-crystallised PC190723 binding site. The amino acid residues and ligand interactions were checked in 2D format for both pre- and post-docking for PC190723 in order to validate that the docking method performed on this co-crystallised site in Maestro [21] was valid and accurate. The ligand interaction diagrams for both pre- and post-docking for PC190723 on FtsZ (PDB ID: 4DXD) were identical to each other. This implied that the docking of compound **3** at this same ligand binding site will provide some useful information on how it compared with PC190723, by showing specific compound–residue interactions at this binding site.

A docking score of −9.164 kcal/mol was found for compound **3** when it was docked into the co-crystallised PC190723 binding site (Figure 9), which is similar to that for PC190723 binding to the same site (−9.458 kcal/mol). Ligand–receptor binding energies were analysed by using molecular mechanics-generalised Born surface area (MM-GBSA) [22], which revealed that the overall binding energy for PC190723 **4** was lower than that for compound **3**, shown in Table 1.

From Figure 10 and Figure 11, it can be seen that hydrogen bonding with residue Asn263 is common to both compound **3** and PC190723. The amine group of the amide in benzamide moiety from PC190723 forms hydrogen bonds with Asn263 and Val207, while hydrogen bonding is also possible between the oxygen from the cyclohexyl carbonyl group of compound **3** and Asn263. Other hydrogen bonding interactions are also present for PC190723 **4**; Leu209 and Gly205 are involved with the carbonyl group from the amide in the benzamide moiety. Hydrogen bonding is also possible between Val310, Asn208 and Leu209 and compound **3**.

A considerable number of favourable hydrophobic contacts were also observed between compound **3** and several amino acid residues, such as Asp199, Asn263, Val297 and Thr309. Specifically, Asn263 and Thr309 interacted with the carbonyl group of the cyclohexyl ring in **3**, whereas Asp199 and Val297 formed numerous hydrophobic interactions with the cyclic ring. Favourable hydrophobic contacts were also seen between PC190723 and the surrounding amino acid residues in the binding site. From these observations, Asn263 seems to be the pivotal amino acid residue involved in establishing favourable chemical interactions such as hydrogen bonding and hydrophobic interactions with the ligand. This molecular docking work has also reinforced our previous hypothesis of the likely importance of the cyclohexyl ring in contributing towards the antibacterial effect observed in compounds **1** and **3**.

## 3. Materials and Methods

### 3.1. Chemistry

#### 3.1.1. General Methods

^1^H and ^13^C NMR spectra were recorded at 400 and 100 MHz, respectively, on a Varian 400-MR magnetic resonance spectrometer with chemical shifts (δ) reported in parts per million (ppm). All spectra were acquired in deuterated solvents and the residual solvent peaks were used as internal references High resolution mass spectrometry was performed on a Bruker 7T Fourier-transform ion cyclotron resonance mass spectrometer (Mass Spectrometry Unit, School of Chemistry, University of Sydney). All starting materials and reagents were commercially available and purchased from Sigma Aldrich (Castle Hill, Sydney, NSW, Australia). Solvents were purchased from Chem-Supply (Gillman, SA, Australia) and were used as received. Flash column chromatography was performed using Reveleris (40–63 µm) Davisil chromatographic silica media. Thin layer chromatography (TLC) was performed using Merck KgaA TLC plates (UV254). A CEM Discover SP microwave reactor system was used for all microwave-assisted reactions. A Symmetry^®^ C18 column, (3.9 × 150 mm, 5 μm particle size, 100 Å, Waters, Milford, MA, USA) and Shimadzu HPLC system, consisting of a DGU-20A5R degassing unit, two LC-30AD pumps, a SIL-30AC autosampler, an SPD-M30A photodiode array (PDA) detector, controlled by CBM-20A communication centre which was connected to a PC with Labsolutions CS software for data acquisition and processing, were used to acquire the HPLC chromatograms.

#### 3.1.2. 6-{(E)-4-Hydroxybenzylidene}-2-[{(E)-3-(4-hydroxyphenyl}acryloyl]cyclohexan-1-one 1 [10]

A suspension of 2-acetylcyclohexanone (2.2 g, 15.6 mmol) and boric anhydride (653 mg, 9.4 mmol) in anhydrous DMF (5 mL) was irradiated (80 °C, 300 W, 250 psi maximum allowed pressure) for 15 min. The resulting red solution, containing the boron-protected intermediate, was treated with the substituted benzaldehyde (4.2 g, 34.4 mmol), additional anhydrous DMF (5 mL), and catalytic *n*-butylamine (50 µL). The reaction mixture was then irradiated under the same conditions for 10 min, which was repeated until TLC indicated no further boron-protected intermediate was present. Monitoring by TLC was performed by checking for the free, unreacted cyclohexanone, which could be obtained by diluting a minute sample of the reaction mixture with methanol and shaken vigorously to cleave the boron complex. The second irradiation step required three 10 min cycles. The cooled resulting mixture was pipetted over a celite plug pre-saturated with n-hexane, and a continuous flow of DCM was eluted through the plug under vacuum until the filtrate became clear. A coloured solid persisted on the surface of the plug consisting of the boron-protected product. The entire plug was then transferred into a round bottom flask and refluxed for 2 h in 1:1 water:acetone (100 mL) to cleave the boron complex. The cooled mixture was filtered through a fresh celite plug, eluted with acetone (20 mL) and the combined filtrate evaporated under reduced pressure to afford the desired product 1 as an orange-red solid, mp 225–227 °C (4.9 g, 90%). Crystals were grown from aq. ethanol. ^1^H NMR (400 MHz, acetone-*d*_6_) δ_H_ 1.77 (2H, m, H-4), 2.69 (2H, t, J = 6.0 Hz, H-3), 2.76 (2H, t, J = 5.6 Hz, H-5), 6.89 (4H, d, J = 8.4 Hz, H-3″, H-5″, H-3‴ and H-5‴), 7.11 (1H, d, J = 15.6 Hz, H-2′), 7.37 (2H, d, J = 8.4 Hz, H-2‴ and H-6‴), 7.61 (3H, m, H-2″, H-6″ and Ar-CH=C), 7.68 (1H, d, J = 15.6 Hz, H-3′), 8.98 (1H, s, Ar-OH), 9.19 (1H, s, Ar-OH); ^13^C NMR (100 MHz, acetone-*d*_6_) δ_C_ 23.8 (CH_2_, C-4), 24.7 (CH_2_, C-3), 27.9 (CH_2_, C-5), 109.1 (quat., C-2), 116.3 (2 × CH, C-3″ and C-5″), 116.8 (2 × CH, C-3‴ and C-5‴), 118.3 (CH, CH=CH-Ar, C-2′), 127.8 (quat., C-1″), 128.5 (quat., C-1‴), 131.2 (2 × CH, C-2″ and C-6″), 131.5 (quat., Ar-CH=C, C-6), 132.8 (2 × CH, C-2‴ and C-6‴), 133.7 (CH, Ar-CH=C, C-4′), 142.7 (CH, CH=CH-Ar, C-3′), 158.8 (quat., C-4‴), 160.7 (quat., C-4″), 179.6 (quat., enol C-O, C-1), 187.1 (quat., ketone C=O, C-1′); X-ray diffraction parameters: Monoclinic; P21/n; a = 8.8357(2) Å, b = 6.9567(2) Å, c = 30.7382(7) Å, α = γ = 90°, β = 92.121(2)° (293 K); Z = 4; R = 0.044; GOF = 1.081. See ESI for ORTEP, refinement data, atom coordinates, and thermal parameters [23,24]. 

#### 3.1.3. 2-(4-Hydroxybenzyl)-6-{3-(4-hydroxyphenyl)propanoyl}cyclohexan-1-one 2 [14]

Cyclohexanone **1** (700 mg, 1.99 mmol) was suspended in excess limonene (20 equiv.) and cooled to 0 °C in an ice-water bath. Then, 10% Pd/C (2.5 mol %) was added slowly before heating to 180 °C for 3 h with a reflux condenser attached. The reaction was quenched by the addition of excess distilled water to the cooled reaction mixture and the resulting mixture was passed through a celite plug under vacuum. The plug was washed with excess methanol, the combined filtrate was concentrated in vacuo, and the majority of the organic solvent was removed by steam distillation in vacuo to obtain a crude residue, which was purified by flash column chromatography (DCM:A—100:0 to 99:1) to afford the desired product **2** as a yellow oil (450 mg, 64%). HPLC: 96.2% (360 nm); IR v_max_/cm^−1^: 3294 (br., Ar-OH), 1612 (C=O); ^1^H NMR (400 MHz, acetone-*d*_6_) δ_H_ 2.95 (2H, t, J = 7.6 Hz, H-3), 3.38 (2H, t, J = 7.6 Hz, H-2), 3.88 (2H, s, Ar-CH_2_-Ar), 6.75 (4H, m, H-3″, H-5″, H-3‴ and H-5‴), 6.85 (1H, m, H-5′), 7.09 (2H, d, J = 8.4 Hz, H-2‴ and H-6‴), 7.13 (2H, d, J = 8.4 Hz, H-2″ and H-6″), 7.34 (1H, d, J = 7.2 Hz, H-4′), 7.85 (1H, d, J = 8.0 Hz, H-6′), 8.12 (2H, s, OH-4″ and OH-4‴), 12.81 (1H, s, OH-2′); ^13^C NMR (100 MHz, acetone-*d*_6_) δ_C_ 29.9 (CH_2_, CH2-CH2-Ar, C-3), 34.7 (CH_2_, Ar-CH_2_-Ar), 41.1 (CH_2_, CH_2_-CH_2_-Ar, C-2), 115.9 (2 × CH, C-3″ and C-5″ or C-3‴ and C-5‴), 116.1 (2 × CH, C-3″ and C-5″ or C-3‴ and C-5‴), 119.4 (CH, C-5′), 119.8 (quat., C-1′), 129.4 (CH, C-6′), 130.1 (quat., C-3′), 130.2 (2 × CH, C-2″ and C-6″), 130.7 (2 × CH, C-2‴ and C-6‴), 131.9 (quat., C-1‴), 132.6 (quat., C-1″), 137.4 (CH, C-4′), 156.6 (2 × quat., C-4″ and C-4‴), 161.1 (quat., phenol C-O, C-2′), 207.7 (quat., ketone C=O, C-1); HRMS (ESI/FT-MS/+ve) *m*/*z*: Calc. for C_22_H_20_O_4_: [M + H]^+^, 349.1440; Found: [M + H]^+^, 349.1434. 

#### 3.1.4. 1-{2-Hydroxy-3-(4-hydroxybenzyl)phenyl}-3-(4-hydroxyphenyl)propan-1-one 3

Cyclohexanone **1** (450 mg, 1.3 mmol) was dissolved and stirred in methanol (10 mL) with 10% Pd/C (1.5 mol%) and cyclohexene (973 mg, 11.8 mmol) was added dropwise. The reaction was heated under reflux for two hours and monitored by TLC. The resulting solution was filtered through celite to obtain the crude product which was purified by flash column chromatography (ethyl acetate:hexane—1:4 to 2:3) to produce **3** as yellow oil (64.9 mg, 15%). HPLC: 96.1% (360 nm); IR ν_max_/cm^−1^: 3350 (OH), 1715 (C=O), 1650 (C=C); ^1^H NMR (400 MHz, methanol-*d*_4_) δ_H_ 2.16–3.35 (12H, m, 2 × H-4′, 2 × H-2′, 2 × H-3′, 2 × H-3, 2 × H-4, 2 × H-5), 3.35 (1H, broad, H-6), 6.69 (4H, m, H-3″, H-5″, H-3‴, H-5‴), 7.00 (4H, m, H-2″, H-6″, H-2‴, H-6‴); ^13^C NMR (100 MHz, methanol-*d*_4_) δ_C_ 29.5 (CH_2_, C-3′), 30.7 (CH_2_, C-2′), 31.4 (CH_2_, C-4′), 35.0 (CH_2_, C-3 or C-4 or C-5), 37.8 (CH_2_, C-3 or C-4 or C-5), 40.3 (CH_2_, C-3 or C-4 or C-5), 41.5 (C-6), 116.1 (4 × CH, C-3″, C-5″, C-3‴, C-5‴), 130.4 (2 × CH, C-2‴, C-6‴ or C-2″, C-6″), 130.9 (2 × CH, C-2″, C-6″ or C-2‴, C-6‴), 132.6 (quat, C-1″ or C-1‴), 133.1 (quat, C-1‴ or C-1″), 156.6 (quat, C-4″ or C-4‴), 161.3 (quat, C-4‴ or C-4″), 208.1 (2 × quat, C-1 and C-1′); HRMS (ESI/FT-MS/ ve) *m*/*z*: Calc. for C_22_H_24_O_4_: [M + Na]^+^, 375.1567; Found: [M+Na]^+^, 375.1564. 

### 3.2. Microbiology

#### 3.2.1. Drug Solution

Compound **3** was dissolved in 100% dimethyl sulfoxide (DMSO), as was compound **1** [10]. The concentration of the drug **3** was calculated from its absorbance at 278 nm with a molar extinction coefficient of 16141 L mol^−1^ cm^−1^.

#### 3.2.2. Determination of Minimum Inhibitory Concentration

The Minimum Inhibitory Concentration (MIC) was determined as described previously, following the agar dilution method [10,25]. From an overnight culture of *Bacillus subtilis* 168 (*Bs* 168), 2% inoculation was performed to a fresh tube containing Luria Broth. When the bacteria reached an O.D_600_ = 0.4, the culture was serially diluted to a dilution factor of 10^4^. Then, 20 μL of the diluted culture was spotted on plates containing different concentrations of 3 (0, 1, 3, 5, 7, 9, 10, 11, 12, 13 μM) and incubated, at 37 °C, for 6 h, after which the colonies were counted. Similar procedure was performed for *Streptococcus pneumoniae* TIGR4. Bacteria were grown in Todd–Hewitt Yeast Broth up to 0.4 OD_600_ and serially diluted. Upon addition on to BHI agar plates containing different concentrations of **3** (0.5, 1, 1.5, 2, 3, 5, 7, 10 μM), the colonies were incubated for 24 h, at 37 °C, with 5% CO_2_. MIC was determined as the concentration of **3** that shows no colonies. IC_50_ was determined by plotting percentage inhibition versus log of concentration of drug, using GraphPad Prism 7 software.

#### 3.2.3. Immunostaining for Z-Ring and Nucleoid

Immunostaining of *Bs* 168 cells was performed as described previously [10,26,27]. Briefly, from an overnight culture of *Bs* 168, 2% inoculation was performed to a fresh tube containing Luria Broth. When the bacteria reached an O.D_600_ = 0.4, the culture was divided into two fractions. To one fraction, 8 μM **3** was added, and to another, an equal volume of DMSO. The cells were incubated at 37 °C under shaking conditions for 10 min. Cells were then fixed for 30 min with 2.8% formaldehyde and 0.04% glutaraldehyde. Cells were permeabilized using 0.1% Triton X-100 and then treated with 1 mg/mL lysozyme and 5 mM EDTA for 45 min. Treatment with 0.5% BSA for 30 min was performed to block non-specific binding sites. Then, 1 h treatment each with primary antibody, anti-FtsZ antibody raised in rabbit (1:50 dilution) and secondary antibody (1:400), Alexa Fluor^TM^ 594 goat anti-rabbit IgG (H+L) was performed. Nucleoid staining was performed with 1 µg/mL 4′-6-diamidino-2-phenylindole (DAPI) for 20 min. Images were captured using Confocal Laser Scanning Microscope—Carl Zeiss, LSM 780. Similar procedure was followed for *Sp* TIGR4, which was treated with 2 μM 3 for 1 h before following the above immunostaining procedure.

#### 3.2.4. Membrane Staining

Membrane staining of live *Bs* 168 cells was performed using FM 4–64 stain [28]. *Bs* 168 cells were grown as described above and treated with 8 μM **3** or equivalent volume of DMSO for 10 min. Cells were pelleted down and resuspended in PBS containing 10 μg/mL FM-4–64 dye. After incubation for another 10 min under shaking conditions in the dark, the cells were washed with PBS to remove excess dye. The cells were then mounted on the slide pre-coated with poly-L-lysine and covered with a coverslip. Images were captured in Confocal Laser Scanning Microscope—Carl Zeiss, LSM 780.

#### 3.2.5. Scanning Electron Microscopy

*Sp* TIGR4 was grown as described above and treated with 2 μM **3** or equivalent volume of DMSO for 1 h. The cells were then fixed with 2.8% formaldehyde and 0.04% glutaraldehyde and resuspended in PBS as described earlier [29]. A small volume (5 μL) was spotted on an aluminium foil and allowed to air dry completely. Images were captured using Field Emission Gun-Scanning Electron Microscope (FEG-SEM) JSM-7600F.

#### 3.2.6. Protein Purification

*Streptococcus pneumoniae* FtsZ (*Spn*FtsZ) and *Bacillus subtilis* FtsZ (*Bs*FtsZ) were purified as described previously [10,30]. Tubulin from goat brain was purified as described previously [31]. The concentration of all the proteins was estimated using Bradford’s reagent [32].

#### 3.2.7. Binding of Compound **3** to SpnFtsZ

The binding of **3** to *Spn*FtsZ was assessed using tryptophan fluorescence [33]. The decrease in the fluorescence intensity of *Spn*FtsZ in the presence of 3 was measured. Then, 2 μM of *Spn*FtsZ in 50 mM PIPES pH 7.4, 50 mM KCl, was incubated with different concentrations of **3** (0.2, 0.5, 1, 1.5, 2, 3, 4, 5, 7, 10), at 25 °C, for 10 min. The fluorescence spectra were measured at an emission range of 310 nm to 410 nm, after excitation at 295 nm in JASCO FP-8300 spectrofluorometer. Due to the absorbance of **3** at 340 nm, inner filter correction was performed using the below equation:(1)$$F_{corrected}=F_{observed}*antilog\;\left(\frac{OD_{ex}+OD_{em}}{2}\right)$$

The binding constant, *K_d_*, was determined using the following equation:(2)$$∆F=\frac{∆F_{max}\times \left[L\right]}{K_{d}+\left[L\right]}$$

Δ*F* is the change in the fluorescence intensity of *Spn*FtsZ, Δ*F*_*max*_ is the maximal change in the fluorescence of *Spn*FtsZ when the receptor site is fully occupied, and [*L*] is the concentration of **3**.

#### 3.2.8. Light Scattering 

*Spn*FtsZ (10 μM) in 50 mM PIPES pH 7.4, 50 mM KCl and 5 mM MgCl_2_ (PKM) buffer was incubated with different concentrations of **3** (0, 1, 2, 3, 5, 6, 7) for 20 min, at 25 °C. After the addition of 1 mM GTP, the polymerization was monitored for 600 s, at 37 °C, using light scattering at 400 nm [30] in a JASCO FP-8300 spectrofluorometer.

#### 3.2.9. Sedimentation Assay

*Spn*FtsZ (10 μM) in PKM buffer was incubated with different concentrations of **3** (0.5, 1, 3, 6, 10) for 20 min at 25 °C. The reaction mixture was allowed to polymerize by adding 1 mM GTP and incubated for 600 s, at 37 °C. The mixture was then subjected to ultra-centrifugation at 60,000 RPM for 30 min at 30 °C. The supernatant was collected and quantified using Bradford’s reagent [32]. The amount of FtsZ present in the pellet was assessed by subtracting the supernatant concentration from the total protein [30].

#### 3.2.10. Transmission Electron Microscopy

*Spn*FtsZ (10 μM) in PKM buffer was incubated without and with 10 μM **3** for 20 min, at 25 °C. The mixture was then polymerized by the addition of 1 mM GTP for 200 s, at 37 °C. Samples for EM were prepared on a fresh carbon–formvar-coated copper EM grid according to a previously described method [34]. Imaging was performed the next day using JEOL JEM 1220 HRTEM (200 kV). ImageJ software was used to determine the length of the FtsZ filaments.

In another experiment, *Bs*FtsZ (5 μM) in PKM buffer was incubated without and with 20 μM **3** for 20 min, at 25 °C. The mixture was then polymerized by the addition of 1 mM GTP for 600 s, at 37 °C. Samples for imaging were prepared using the same procedure described above.

#### 3.2.11. Atomic Force Microscopy

Then, 4 μM *Spn*FtsZ in PKM buffer was incubated with or without 10 μM **3**, for 20 min, at 25 °C. Upon addition of 1 mM GTP, FtsZ was polymerized for 300 s, at 37 °C. AFM samples were prepared according to a previously described method [35]. Then, 20 μL of each sample was added onto the mica sheet and allowed to absorb on it for 15 min. The sheet was then washed with MilliQ and allowed to dry overnight. The samples were imaged the next day using MFP-3D BIO, Asylum Research USA.

#### 3.2.12. Tubulin Polymerization Assay

Tubulin polymerization was performed as described previously [36]. Purified tubulin (13 μM) in PEM buffer (50 mM PIPES pH 6.8, 3 mM MgCl_2_ and 1 mM EGTA) + 10% DMSO was incubated with **3** for 10 min on ice. Then, 1 mM GTP was added to the mixture, and polymerization was monitored by light scattering in SoftMax Spectra Multi Plate reader for 30 min, at 37 °C. The excitation and emission wavelengths were set to 350 nm.

#### 3.2.13. Evaluation of the Activity of Alkaline Phosphatase

Alkaline phosphatase assay was performed as described previously [36]. Then, 2 U/mL Alkaline Phosphatase enzyme in glycine NaOH buffer, pH 10.4 with 100 mM glycine, 1 mM MgCl_2_, and 0.1 mM ZnCl_2_ was incubated with different concentrations (5, 10, and 20 μM) of **3** for 10 min on ice. Immediately after the addition of 250 μM para-4-nitrophenyl phosphate (PNPP), the absorbance at 410 nm was measured for 700 s in a JASCO V-730 spectrophotometer.

#### 3.2.14. Statistical Analyses

Statistical analyses were performed using Graph-Pad Prism online software (GraphPad QuickCalcs Website: https://www.graphpad.com/quickcalcs/ttest1/ accessed on 28 February 2019) for the unpaired *t*-test. The differences were considered non-significant if the *p* values were more than 0.05.

## 4. Conclusions

In this study, we have demonstrated the anti-bacterial potential of enol **3** against the pathogen, *S. pneumoniae*, which is mainly responsible for meningitis and community-acquired pneumonia (CAP). *S. pneumoniae* resistance to penicillins and macrolides is increasing, highlighting the need for new anti-pneumococcal agents [37]. We identified that the enol **3** acts by hindering the cell division process by perturbing Z-ring dynamics inside the cell. Enol **3** was also shown to inhibit FtsZ polymerization and induce its aggregation in vitro. We ruled out the possibility that enol **3** induced non-specific inhibition of protein molecules by showing that it does not affect the activity of tubulin and alkaline phosphatase. Docking studies show that enol **3** binds near the T7 loop, which is the catalytic site of FtsZ. A possible explanation could be that **3** binding to the catalytic site of FtsZ interferes with amino acid interactions, leading to accelerated aggregate formation of FtsZ. These large aggregates may not allow further FtsZ polymerization, thereby leading to disruption of the dynamic Z-ring, as seen in the cells. Similar effects on Z-ring and FtsZ assembly were also observed in *B. subtilis*, indicating that enol **3** could be a broad-spectrum anti-bacterial agent useful in targeting Gram-positive bacteria.

In conclusion, enol **3** shows strong anti-pneumococcal activity as a result of its effect upon Z-ring dynamics, prompting further pre-clinical studies to explore its potential as a treatment for bacterial infections.

## Figures and Tables

**Figure 1 molecules-27-06993-f001:**
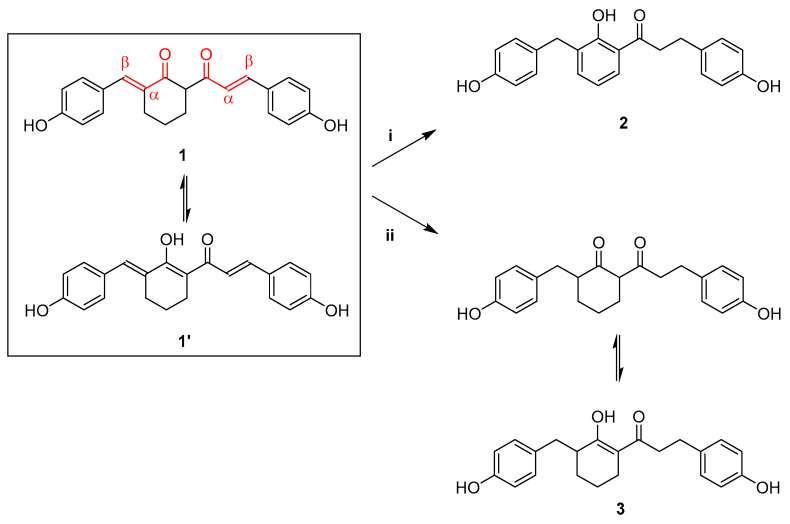
Removal of enone functionalities in cyclohexanone derivative **1** through disproportionation to phenol **2** or catalytic transfer hydrogenation to reduced 1,3-diketone **3**. Reagents and conditions: (i) 10% Pd-on-C, *d*-limonene, 0 °C, then 180 °C, 3 h, 64%; (ii) 10% Pd-on-C, cyclohexene, methanol, 65 °C, 2 h, 15%.

**Figure 2 molecules-27-06993-f002:**
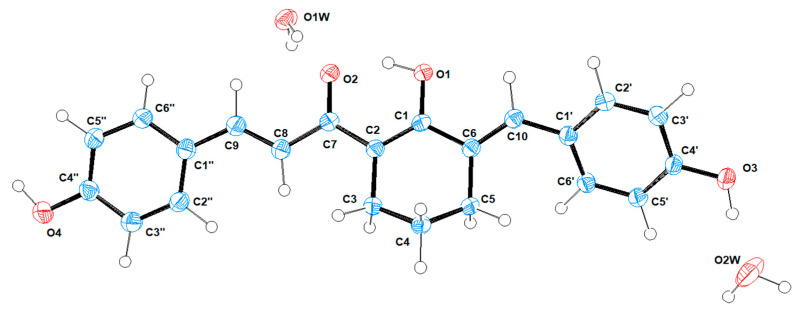
Oak-Ridge Thermal Ellipsoid Plot (ORTEP) of the Independent Atomic Model (IAM) of compound **1**, which crystalised from aq. ethanol in the enol form **1′**. Data collected at 150K, monoclinic (P2_1_/n), goodness-of-fit on F^2^ = 1.078, R_1_% = 0.0393. Solved using intrinsic phasing, (SHELX-T), refined using Least Squares (SHELX-L) [16]. For full crystallographic details see ESI.

**Figure 3 molecules-27-06993-f003:**
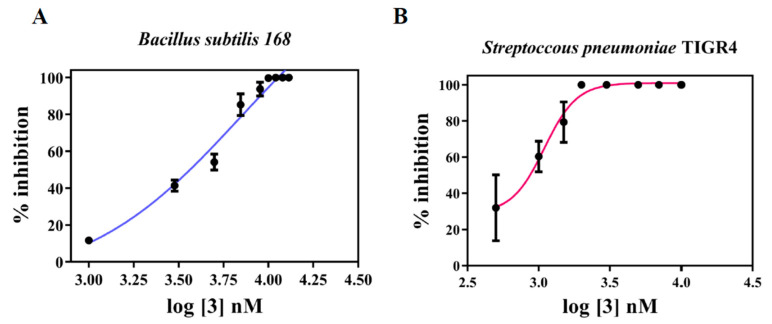
Compound **3** shows strong anti-bacterial activity. The percentage inhibition of proliferation was plotted against the log concentration of reduced enol **3** against (**A**) *Bacillus subtilis* 168, (**B**) *Streptococcus pneumoniae* TIGR4. Error bars represent the standard deviation from three independent sets of experiments.

**Figure 4 molecules-27-06993-f004:**
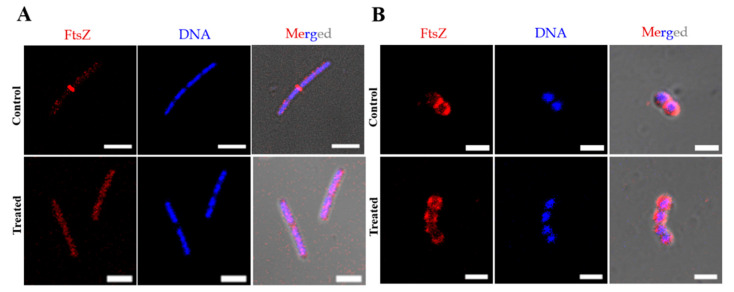
Enol **3** causes Z-ring delocalization in *B. subtilis* and *S. pneumoniae* without affecting the DNA segregation. Fluorescence (Alexa Fluor 594 and DAPI channels for FtsZ and DNA, respectively) and DIC-merged laser scanning confocal microscopy images of fixed cells. (**A**) *B. subtilis* 168 cells immunostained after treatment with and without 8 μM **3** for 10 min. The cells shown are representative of 600 cells imaged and counted in each case. Scale bar indicates 5 μm. (**B**) *S. pneumoniae* TIGR4 cells immunostained after treatment with, or without, 2 μM **3** for 1 h. The cell is representative of 400 cells imaged and counted in each case. Scale bar indicates 2 μm.

**Figure 5 molecules-27-06993-f005:**
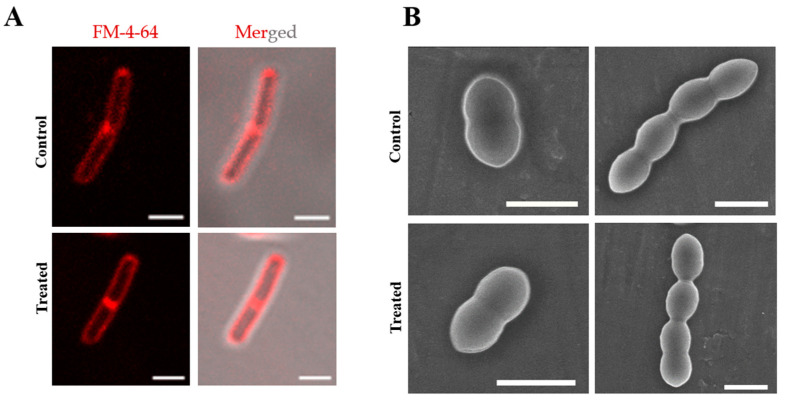
Enol **3** has no effect on the membrane or morphology of *B. subtilis* and *S. pneumoniae* cells. (**A**) Fluorescence (FM-4-64 channel) and DIC-merged laser scanning confocal microscopy images of *Bacillus subtilis* 168 cells stained for membrane using FM-4-64 dye after treatment with and without 8 μM **3** for 10 min. The images are representative of 300 cells imaged in each case. Scale bar indicates 2 μm. (**B**) Scanning electron microscopy images of fixed *S. pneumoniae* TIGR4 cells treated with, or without, 2 μM **3** for 1 h. The cells shown are representative of 100 cells imaged in each case. Scale bar indicates 2 μm.

**Figure 6 molecules-27-06993-f006:**
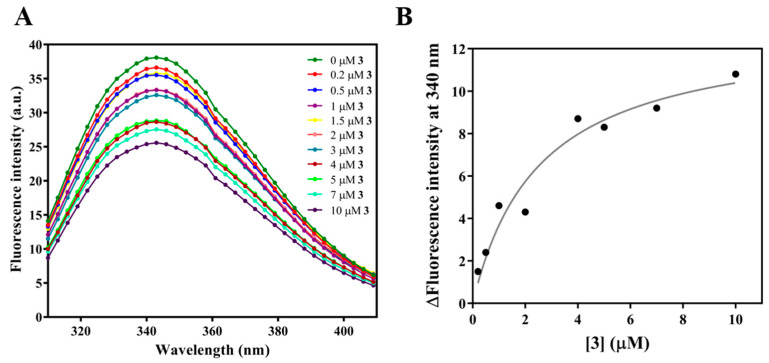
Compound **3** binds to FtsZ with strong affinity. (**A**) Representative spectra of tryptophan fluorescence of *Streptococcus pneumoniae* FtsZ (*Spn*FtsZ) in the presence of increasing concentrations of **3**. *Spn*FtsZ was excited at 295 nm. (**B**) Three independent sets of experiments are represented in the plot of change in fluorescence intensity at 340 nm vs. concentration of **3**, which was used to determine the binding constant (K_d_).

**Figure 7 molecules-27-06993-f007:**
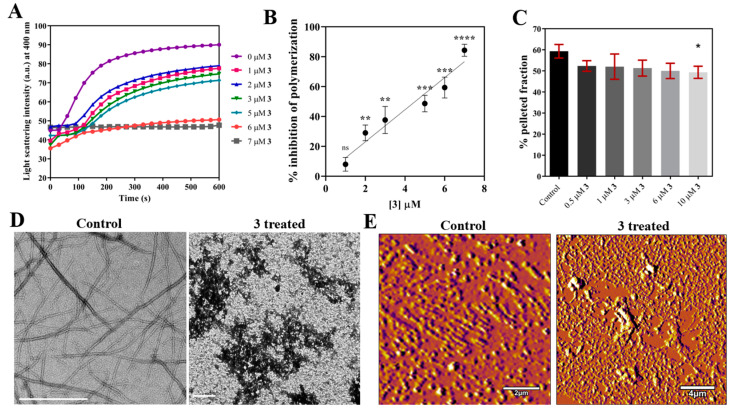
Compound **3** inhibits FtsZ polymerization and induces FtsZ aggregation. (**A**) *Spn*FtsZ (10 μM) in 50 mM PIPES pH 7.4, 50 mM KCl and 5 mM MgCl_2_ (PKM) buffer was incubated with different concentrations of **3** for 20 min at 25 °C polymerized in the presence of 1 mM GTP. The polymerization was monitored using light scattering with emission and excitation wavelength set to 400 nm. (**B**) The change in the extent of light scattering representing *Spn*FtsZ polymerization in the presence of **3** was plotted as percentage inhibition Vs concentration of **3**. The error bars indicate standard deviation from three independent sets of the experiment. (^ns^ indicates non-significant with *p* ≥ 0.05, ** indicates *p* ≤ 0.01, *** indicates *p* ≤ 0.001, **** indicates *p* ≤ 0.0001.) (**C**) Using the same conditions mentioned above, *Spn*FtsZ was polymerized and subjected to ultra-centrifugation to separate the pellet and the supernatant fraction. The percentage of pellet fraction is plotted against the concentration of **3**. The error bars indicate standard deviation from three independent sets of the experiment. (* represents *p* = 0.02.) (**D**) Transmission electron microscopy images of *Spn*FtsZ filaments formed in the presence and absence of **3**. *Spn*FtsZ (10 μM) in PKM buffer was incubated without and with 10 μM **3** for 20 min, at 25 °C, before polymerizing in the presence of 1 mM GTP. Scale bar indicates 1 μm. This experiment was performed three times. (**E**) Atomic force microscopy (AFM) images of *Spn*FtsZ filaments formed in the presence and absence of 10 μM **3**. Here, 4 μM *Spn*FtsZ was polymerized using the same conditions mentioned above.

**Figure 8 molecules-27-06993-f008:**
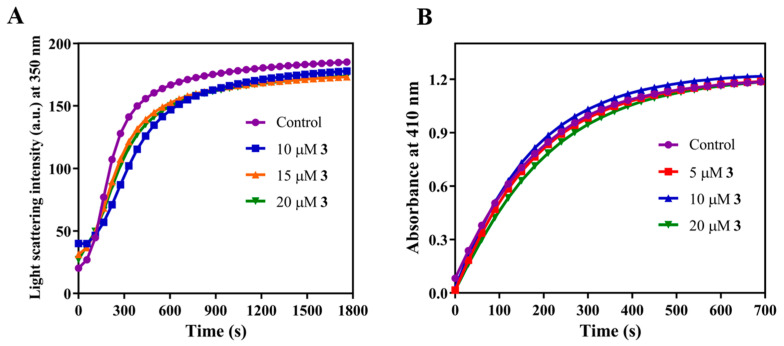
Compound **3** does not inhibit the activity of tubulin and alkaline phosphatase. (**A**) Mammalian tubulin (13 μM) in PEM buffer was polymerized after incubation with increasing concentration of **3** and monitored using light scattering at 350 nm. The spectra are representative of three independent sets of experiment. (**B**) The activity of Alkaline phosphatase incubated with increasing concentrations of **3** was monitored using absorbance at 410 nm after addition of PNPP substrate. The spectra are representative of three independent sets of experiment.

**Figure 9 molecules-27-06993-f009:**
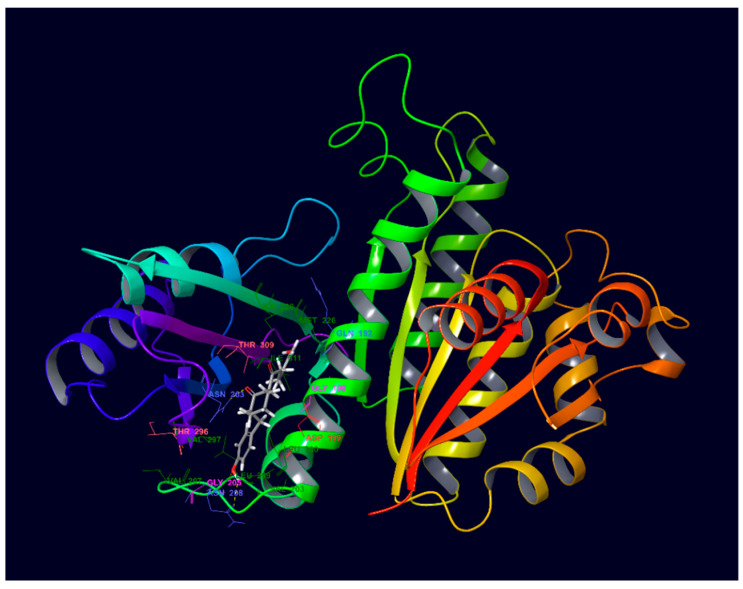
Compound **3** docked into PC190723 **4** binding site on FtsZ (PDB ID: 4DXD).

**Figure 10 molecules-27-06993-f010:**
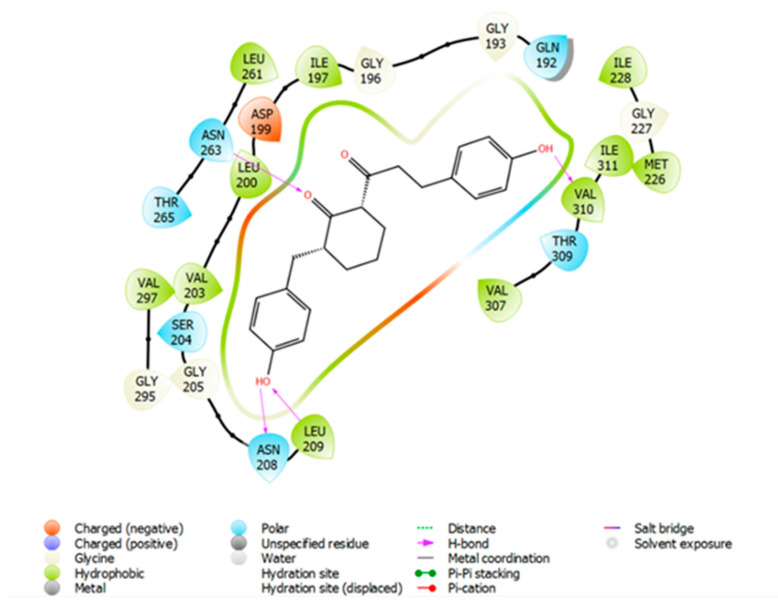
A 2D view of compound **3** docked into the PC190723 **4** binding site of FtsZ (PDB ID 4DXD).

**Figure 11 molecules-27-06993-f011:**
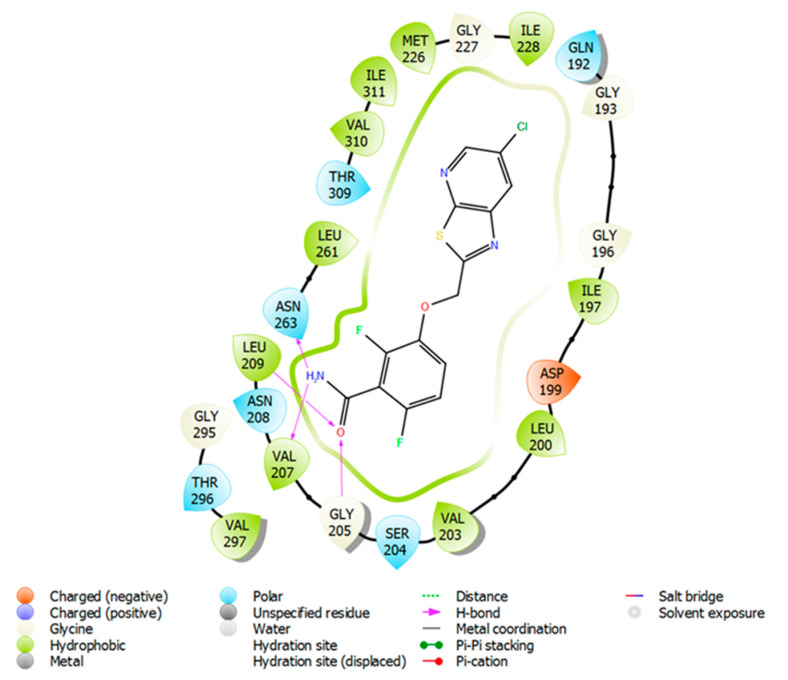
A 2D view of PC190723 **4** docked into crystal structure of PC190723 **4** bound to FtsZ (PDB ID 4DXD).

**Table 1 molecules-27-06993-t001:** MM-GBSA binding energies of compound **3** and PC190723 **4** on FtsZ (PDB ID: 4DXD).

	MM-GBSA ΔG Binding Energy (kcal/mol)	Ligand Strain Energy (kcal/mol)	Receptor Strain Energy (kcal/mol)
Compound **3**	−56.53	9.43	13.38
PC190723 **4**	−71.39	2.42	7.71

## Data Availability

All data is available from the authors.

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
