# Peer review of "Discovery of 2′,6-Bis(4-hydroxybenzyl)-2-acetylcyclohexanone, a Novel FtsZ Inhibitor"

_molecules, 2022, doi:10.3390/molecules27206993_

Round 1

Reviewer 1 Report

The paper is interesting and well organized, the methodology is clear and the results are significant.

In order to overcome possible sites of degradation/metabolism on compound 1, while hopefully maintaining the desired antibacterial activity, you succeeded to found compound 3 with a similar antibacterial activity against Bs168 cells and very potent against the pathogen S. pneumoniae, which is a great result.

 The abstract is very well written, precise and clear.

The literature is scarce, there are many new papers dealing with FtsZ inhibitors, which have not been mentioned, in this sense the literature should be expanded and modernized.

In the following sentence (lines 41-43): We have recently reported that a non-natural curcumin analog 1 possesses powerful antibacterial activity against important pathogenic bacteria, as a result of its targeting of FtsZ, literature citation is missing.

The results are fine, but the introduction is unclear and confusing, it may be reformulated, in order that compound 1 is not generally emphasized so much if it is possible.

Author Response

Thank you for your comments.

The literature is scarce, there are many new papers dealing with FtsZ inhibitors, which have not been mentioned, in this sense the literature should be expanded and modernized.

Response - we have included 3 recent reviews of FtsZ inhibitors to provide greater background to the development of these compounds.

In the following sentence (lines 41-43): We have recently reported that a non-natural curcumin analog 1 possesses powerful antibacterial activity against important pathogenic bacteria, as a result of its targeting of FtsZ, literature citation is missing.

Response - we have included the missing reference

The results are fine, but the introduction is unclear and confusing, it may be reformulated, in order that compound 1 is not generally emphasized so much if it is possible.

Response - We have reduced the emphasis on compound 1 in the introduction and on the PAINs.

Reviewer 2 Report

The manuscript that is presented is excellent, the figures are clear and the language is very well understood, I recommend its publication.

Author Response

Thank you for your comments.

In response to the editor's comments, we have moved the materials and methods into the main document, as well as the 2 figures requested and the table.

Reviewer 3 Report

Lin and co worker have successfully shown that the enol compound 3 acts as a potent anti-pneumococcal agent  by hindering the cell division process by perturbing Z-ring dynamics inside the cell. The compound 3 was also shown to inhibit FtsZ polymerization and induce its aggregation in vitro but does not affect the activity of tubulin and alkaline phosphatase. I recommend acceptance of the manuscript after addressing the following points.

1. please provide melting point of the solid compound 1

2. Please add N values at the figure legends.

3. please specify the version of GraphPad prism was used for statistical analysis.

Author Response

Thank you for your comments.

1 The melting point of compound 1 has been included (line 353)

2 The number of repeats for each of the biological evaluations is given in the legend under each figure

3 The version of Graph Pad Prism has been included (line 423)